# Genome Size Variation within Species of Chinese Jujube (*Ziziphus jujuba* Mill.) and Its Wild Ancestor Sour Jujube (*Z. acidojujuba* Cheng et Liu)

**Lihu Wang [1,2], Zhi Luo [2,3], Zhiguo Liu [1,2], Jin Zhao [4], Wenping Deng [5], Hairong Wei [5], Ping Liu [1,2] and Mengjun Liu [1,2,\*]**

1   Research Center of Chinese Jujube, Hebei Agricultural University, Baoding 071001, China; hdwanglh@163.com (L.W.); jujubeliu@163.com (Z.L.); yylp@hebau.edu.cn (P.L.)
2   Jujube Industry Technology Research Institute of Hebei, Baoding 071001, China; lz15930762701@126.com
3   College of Forestry, Hebei Agricultural University, Baoding 071001, China
4   College of Life Sciences, Hebei Agricultural University, Baoding 071001, China; zhaojinbd@126.com
5   School of Forest Resources and Environmental Science, Michigan Technological University, Houghton, MI 49931, USA; Wenpingd@mtu.edu (W.D.); hairong@mtu.edu (H.W.)
\*   Correspondence: kjliu@hebau.edu.cn; Tel.: +86-139-3226-2298

**Abstract:** One of the most important attributes of a genome is genome size, which can to a large extent reflect the evolutionary history and diversity of a plant species. However, studies on genome size diversity within a species are still very limited. This study aims to clarify the variation in genome sizes of Chinese jujube and sour jujube, and to characterize if there exists an association between genome sizes and geographical variation. We measured the genome sizes of 301 cultivars of Chinese jujube and 81 genotypes of sour jujube by flow cytometry. Ten fruit traits, including weight, vertical diameter, horizontal diameter, size, total acids, total sugar, monosaccharide, disaccharide, soluble solids, and ascorbic acid were measured in 243 cultivars of Chinese jujube. The estimated genome sizes of Chinese jujube cultivars ranged from 300.77 Mb to 640.94 Mb, with an average of 408.54 Mb, with the highest number of cultivars (20.93%) falling in the range of 334.787 to 368.804 Mb. The genome size is somewhat different with geographical distribution. The results showed weakly significant positive correlation ($p < 0.05$) between genome size and fruit size, vertical diameter, horizontal diameter, and weight in the Chinese jujube. The estimated sour jujube genome sizes ranged from 346.93 Mb to 489.44 Mb, with the highest number of genotypes (24.69%) falling in the range of 418.185 to 432.436 Mb. The average genome size of sour jujube genotypes is 423.55 Mb, 15 Mb larger than that of Chinese jujube. There exists a high level of variation in genome sizes within both Chinese jujube cultivars and sour jujube genotypes. Genome contraction may have been occurred during the domestication of Chinese jujube. This study is the first large-scale investigation of genome size variation in both Chinese jujube and sour jujube, which has provided useful resources and data for the characterization of genome evolution within a species and during domestication in plants.

**Keywords:** genome size; variation; expansion; contraction; flow cytometry; Chinese jujube; sour jujube

## 1. Introduction

Genome size is considered to be one of the important attributes of a genome. It can, to a large extent, reflect the evolution history [1] and biodiversity [2] of a plant species. It has been demonstrated that genome sizes may influence where, when, and how plants grow and have considerable ecological significance [3]. Recent data on the genome sizes of 15,000 eukaryotic species have manifested an astonishing range, varying over 64,000-fold [4], with the genome sizes of 12,000 land plant species

varying over 2400-fold. Within the same species, the variation of genome sizes is also ubiquitously present—for instance, in *Artemisia annua* L. [5], orchids [6], and *Knautia* [7]. Previous studies have shown that plant genome size variation, to a large degree, aligns well with changing environments that may affect either the adaptation and flavors and nutrients [8].

Feulgen microdensitometry was routinely used for plant genome size estimation in the past, but it is laborious and time-consuming [3]. With the availability of DNA flow cytometry (FCM), genome size estimation has become increasingly accessible and accurate, because FCM facilitates more rapid analysis of a larger number of samples [9]. At the same time, FCM has also been applied to the identification of plant ploidy levels [10–13]. FCM analysis has revealed a high level of variation in genome size among species, but the studies on genome size variation within a species are still very limited.

Chinese jujube (*Ziziphus jujube* Mill.), also known as the Chinese date, is an important fruit tree (2n = 2x = 24) in the family of Rhamnaceae. The fruit active ingredient of Chinese jujube has recently been affirmed to possess strong anti-cancer activity [14,15], and is especially rich in ascorbic acid, flavonoid, amino acids, mineral constituents, cerebrosides, and cyclic adenosine monophosphate [16], in addition to carbohydrates, making it highly valuable in both nutrition and medication. It has been cultivated in China for up to 7,000 years, and has been introduced to 47 countries throughout the Americas, southern and eastern Asia, Europe, and Australia [17]. Sour jujube (*Z. acidojujuba* Cheng et Liu, 2n = 2x = 24), a well-known traditional medicinal plant, is generally considered to be the direct wild ancestor species of Chinese jujube. Previous studies have revealed that almost all types of Chinese jujube can find their ancestor in sour jujube [18]. Both Chinese jujube and sour jujube have large variations in tree height, fruit size, fruit nutrient, fruit coloration, fruit flavor, and leaf size [19], which may be partially attributable to different genome sizes. We have successfully established a method for measuring genome sizes of Chinese jujube and sour jujube by flow cytometry [20], which enables us to achieve large-scale estimation of the genome sizes of the two species.

In this study, we measured the genome sizes of 301 cultivars of Chinese jujube and 81 genotypes of sour jujube by flow cytometry. We subsequently compared the differences in genome size among species and distribution. This study yielded valuable data resources and information for a better understanding of genome size variation within the two species, in order to delve into the evolutionary aspects and facets of Chinese jujube and sour jujube, which will help pave the way toward genetic improvements and cultivation of these species for benefiting human health.

## 2. Materials and Methods

### 2.1. Plant Materials

The 301 cultivars of Chinese jujube and the 81 genotypes of sour jujube were sampled at the National Jujube Germplasm Repository (NJGR), Taigu county, Shanxi Province, China, and at the Agricultural Experiment Station of Hebei Agricultural University (HAU), Baoding, Hebei Province, China, respectively. The Chinese jujube cultivars conserved at NJGR were collected from different provinces of China, where they have been cultivated for hundreds or thousands of years, and sour jujube genotypes conserved at HAU were collected from their wild habitats in various provinces of China. Supplemental Table S1 listed all the test materials and their origins. Supplemental Table S2 listed the climatic conditions and climate types of major Chinese provinces.

A Chinese jujube cultivar "Dongzao" (in vitro culture tissues), whose genome has been sequenced (444 Mb) [21], was obtained from the Research Center of Chinese Jujube, Hebei Agricultural University, Hebei province, China. *Populus trichocarpa* (in vitro culture tissues), the other species whose genome has been sequenced (480 Mb), was acquired from the Key Laboratory of Forest Germplasm Resources and Forest Protection, Hebei province, China. *P. trichocarpa* was used as the baseline reference for measuring the genome size of Chinese jujube and sour jujube, in order to ensure the reliability of

the measured results. Three independent repeats in each cultivar or genotype were measured for estimating average genome size and standard deviation.

## 2.2. Flow Cytometry Measurement

Sample collection and processing were conducted according to our previously established methods [20]. Nuclear suspensions were prepared as follows: fresh young leaves from each selected tree were selected and thoroughly rinsed with distilled water three times, and were then chopped with a single front razor blade before they were transferred into tubes containing 2 mL Tris–MgCl$_2$ (200 mmol/L Tris, 4 mmol/L MgCl$_2$·6H$_2$O, 0.5% (*v/v*) TritonX-100, pH 7.5) buffer, which was then incubated at 4 °C for 5 min. The leaf materials were filtered through a green Partec filter (40 μm) and stained with 30 μL propidium iodide (1 mg/mL) and 1 μL RNase (10 mg/mL).

Nuclear suspensions were analyzed by flow cytometry (Partec, Cube8, Germany) equipped with a blue 488 nm laser at a low flow rate (0.5 μL/s). Each sample with at least 10,000 nuclei was collected before it was subjected to FCM analysis. A histogram of DNA content of each sample was evaluated using the Sysmex Express 4 Flow Research Edition software. Only data collected from samples that had G1/G0 (pre-synthesis of DNA in interphase in cell division) peaks with a coefficient of variation (CV) <5% were used to estimate the samples' genome sizes. Three independent repeats in each cultivar or genotype were measured for the purpose of estimating average genome size and standard deviation.

The genome size was estimated according to the formula below:

$$\text{sample genome size (Mb)} = 480 \times \frac{\text{sample G0/G1 peak mean}}{\textit{Populus trichocarpa} \text{ G0/G1 peak mean}} \tag{1}$$

## 2.3. Fruit Morphological Analysis

Fruit samples of 243 cultivars of Chinese jujube were harvested at the ripening stage from three different trees for each cultivar. Ninety representative fruits were sampled from each tree. Fruit morphological traits, including fruit weights (FW), fruit vertical diameter (FVD), fruit horizontal diameters (FHD), and fruit sizes (FS), were determined as described in a previous study [22]. The fruit sizes were estimated by volume.

## 2.4. Fruit Chemical Analysis

A total of 243 cultivars of Chinese jujube under stable growth condition for at least 10 years were selected from 301 cultivars for the analysis of fruit chemical traits, including total acids (TA), total sugar (TS), monosaccharide (MA), disaccharide (DA), soluble solids (SS), and ascorbic acid (Vc). These fruit traits were determined by following the procedure, as described previously [22].

## 2.5. Statistical Analyses

The data were analyzed by Excel (Microsoft Office, 2018) and the figures were made using a R package called ggplot2 and the Performance Analytics (correlation analysis and visualization) in the R statistical software (R Development, Core Team 2018). The Pearson product–moment correlation in the Performance Analytics package was used for correlation analysis. A Student's *t*-test was used to compare the difference between two groups of data and evaluate the level of statistically significant difference between them.

## 3. Results

### 3.1. Validation of the Accuracy of the Genome Size Estimation Method

In order to ensure the reliability of the measured results, *Z. jujuba* "Dongzao" (in vitro culture), whose genome (444 Mb) was sequenced in 2014 [21], was used as a benchmark for measuring the genome size of *P. trichocarpa*. The sample plant was determined by using *P. trichocarpa* as the reference

plant in this study. For both *P. trichocarpa* and "Dongzao", striking peaks with low CV values and low background noise were obtained (Figure 1).

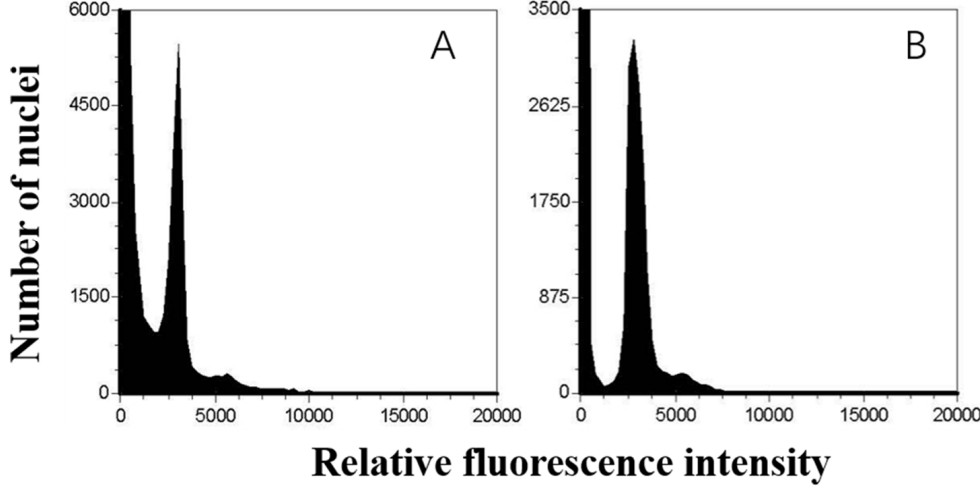

**Figure 1.** Histograms of the DNA contents of *Populus trichocarpa* (**A**) and *Ziziphus jujuba* "Dongzao" (**B**).

The estimated genome size of "Dongzao" is 444.27 Mb (Table 1), which has only a 0.06% difference compared with the sequenced genome size [21]. This result corroborated that the method is reliable for measuring the genome sizes of different species.

**Table 1.** Genome sizes of *Z. jujuba* "Dongzao" and *P. trichocarpa*.

| Cultivar/Species | Relative Fluorescence Density | Coefficient of Variation (%) | Predicted Genome Size (Mb) |
|---|---|---|---|
| *Populus trichocarpa* | 3238.96 ± 96.89 | 3.25 | 480 (reference) |
| Dongzao | 2997.07 ± 50.44 | 4.03 | 444.27 ± 6.31 |

*3.2. Genome Size Variation within Chinese Jujube*

The genome sizes of 301 cultivars of Chinese jujube were measured. The results showed that there was a considerable variation in the genome size of Chinese jujube (Supplemental Table S3). The estimated Chinese jujube genome size ranged from 300.77 Mb to 640.94 Mb, with an average of 408.54 Mb. Among these cultivars, "Zanhuangdazao", "Zanxindazao", and "Jinzandazao", which had much larger genome sizes (640.94 Mb, 626.21 Mb, and 627.13 Mb, respectively), had been identified as triploid in a previous study [19]. When they were excluded, the rest of the cultivars had an average genome size of 406.29 Mb. On the contrary, "Wutaimuzao" has the smallest genome size (300.77 Mb).

The genome sizes of the 301 cultivars were plotted in a histogram where there were 10 bins (groups) (Figure 2). The number of cultivars falling into each bin/group was 42, 40, 63, 56, 53, 27, 15, 2, 0, and 3, respectively. The genome sizes in the group with highest number of cultivars (20.93%) ranged from 334.787 to 368.804 Mb.

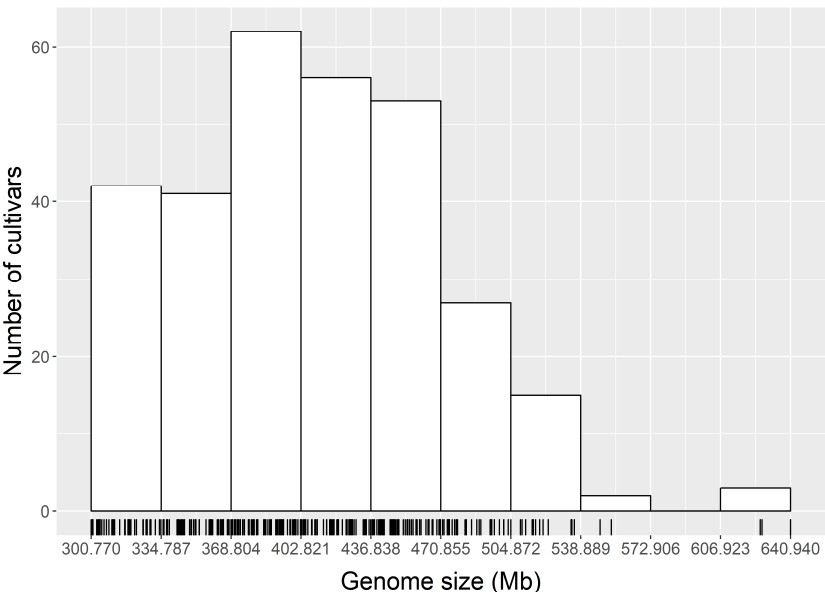

**Figure 2.** Histogram distribution of genome sizes of 301 cultivars of Chinese jujube.

*3.3. Geographical Variation in the Genome Size of Chinese Jujube*

A boxplot was made to manifest the variation of genome size in each province of Chinese jujube, as shown in Figure 3. The results indicate a striking variation that existed in the genome sizes of Chinese jujube cultivars across provinces, especially in those from traditional production areas like Hebei, Henan, Shandong, Shanxi, and Shaanxi. The results also showed that the average genome sizes of Chinese jujube cultivars in different provinces were quite different.

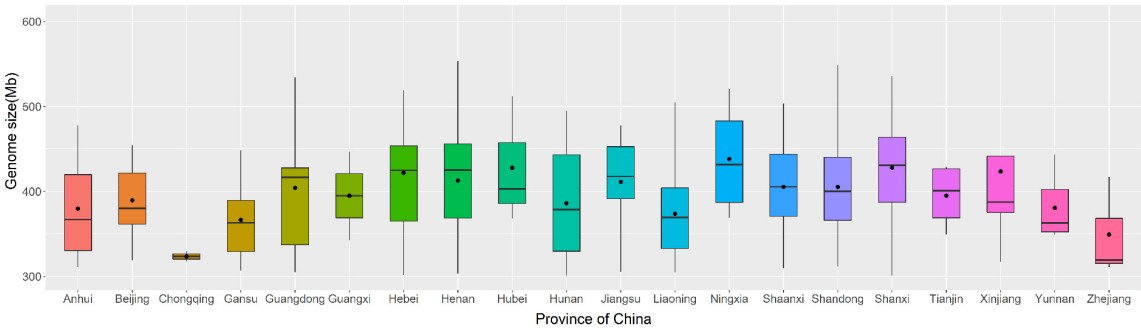

**Figure 3.** Boxplot of genome size variations of Chinese jujube in different provinces. The horizontal black line within each bar represents the median value of the genome size in a province, while a black dot within the bar denotes the average value of genome size in a province.

To visualize the genome sizes of different provinces in relation to the geographical distribution of these provinces, we generated Figure 4, where the number of samples in each province was at least three. The average genome sizes of the traditional production areas that include, but are not limited to, Hebei, Henan, Shandong, Shanxi, and Shaanxi provinces were larger than 400 Mb. The average genome sizes of Chinese jujube for Liaoning province, located in northern China, and Gansu, located in western China, were 373.46 Mb and 366.34 Mb, respectively.

The measured genome size of each cultivar is displayed on the map of China at its geographic location, as shown in Figure 5. The results indicate that genome sizes do not, to a large degree, change in the longitudinal or latitudinal dimensions. The cultivars whose genome sizes are more than 400 Mb were distributed primarily in the traditional production provinces of Chinese jujube, such as Hebei, Shanxi, and Shaanxi. It is noteworthy that the cultivars whose genome size are less than 400 Mb

were distributed largely in the periphery of the traditional distribution area, where one or multiple environmental variables may constitute a limiting factor for Chinese jujube. For example, Liaoning is northeast of Hebei, and is much colder in winter than Hebei. Gansu and Xinjiang are both arid areas with meager precipitation, while Chongqing and Jiangsu, as well as Hunan, are very humid and hot in the summer.

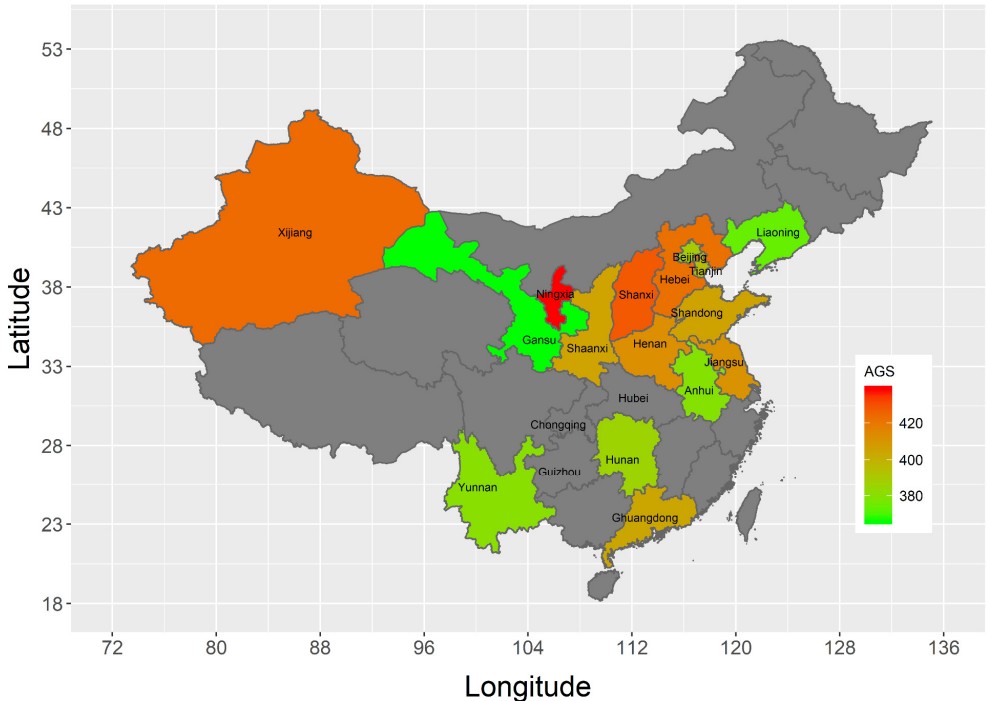

**Figure 4.** A map illustrating the average genome sizes of Chinese jujube cultivars in different provinces of China. AGS: Average genome size (Mb), indicated by the colors shown.

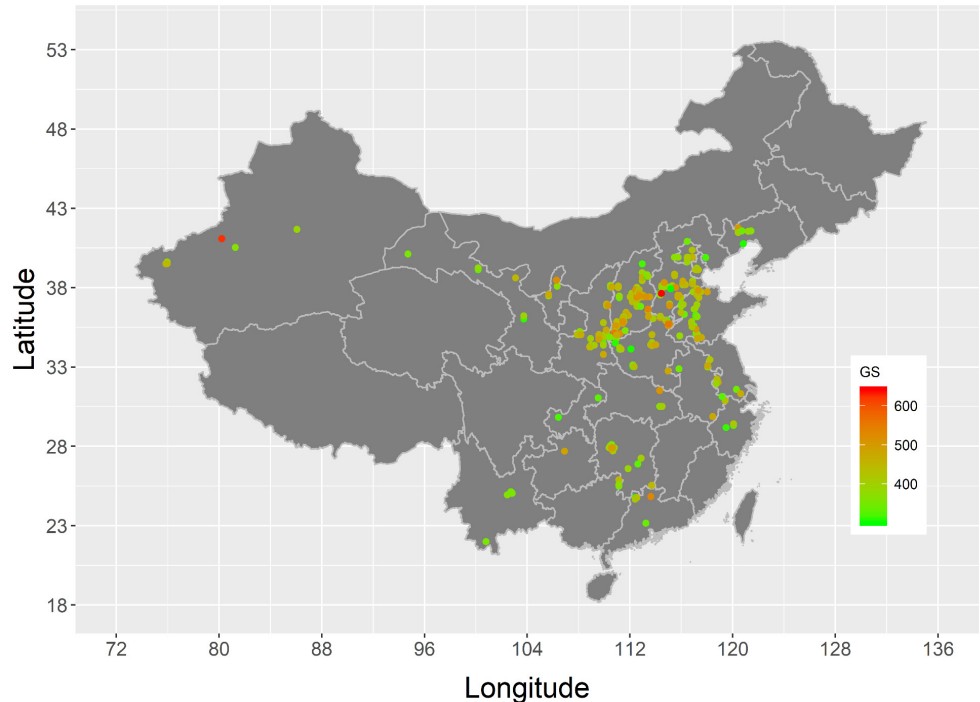

**Figure 5.** Geographic distribution of genome sizes of Chinese jujube cultivars. GS: genome size (Mb). The names and areas of the province correspond to the same areas shown in Figure 4.

*3.4. Morphological and Chemical Variation in Fruit among Cultivars of Chinese Jujube*

The fruit weights (FW), fruit vertical diameters (FVD), fruit horizontal diameters (FHD), and fruit sizes (FS) of 243 cultivars of Chinese jujube were measured (Supplemental Table S3). The results showed considerable changes exist in the four fruit traits between different cultivars. The ranges of FW, FVD, FHD, and FS were 2.48g to 25.08g, 1.9cm to 5cm, 1.4cm to 3.7cm, and 3.54cm$^3$ to 47.82cm$^3$, respectively, and the means of FW, FVD, FHD, and FS were 10.43g, 3.42cm, 2.50cm, and 17.61cm$^3$, respectively.

The FW, FVD, FHD, and FS of 243 cultivars were plotted individually in a histogram where there were 10 bins (groups). As showed in Figure 6, the bins in the histograms of FW, FVD, FHD, and FS that had the highest numbers of cultivars ranged from 7.00g to 9.26g, 3.45 to 3.76cm, 2.09 to 2.32, and 12.396 to 16.824, respectively.

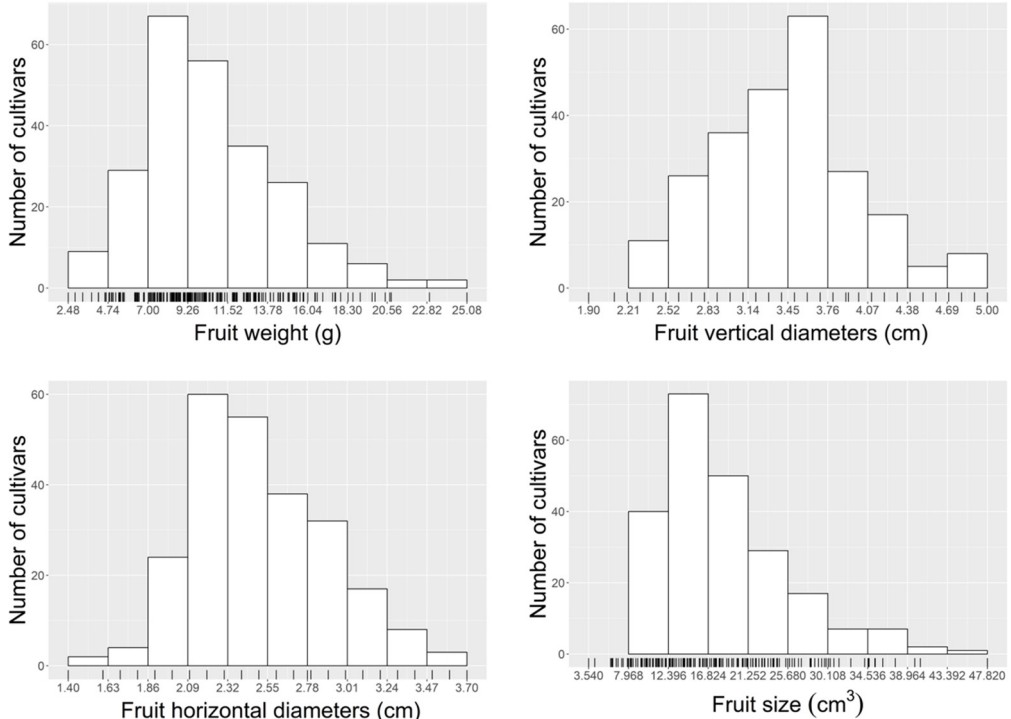

**Figure 6.** Histogram distribution of four fruit morphological traits of 243 Chinese jujube cultivars.

The fruit total acids (TA), total sugar (TS), monosaccharide (MA), disaccharide (DA), soluble solids (SS), and ascorbic acid (Vc) of 243 cultivars of Chinese jujube were also measured (Table S4). Obvious variations among cultivars were found in the six fruit chemical traits. The ranges of TA, TS, MA, DA, SS, and Vc were from 0.31% to 1.69%, 14.61% to 35.41%, 5.15% to 31.18%, 0% to 20.41%, 17% to 43%, and 32.89 to 708.00 mg/100g, respectively. The average values of these six fruit chemical traits were 0.71%, 22.52%, 12.86%, 9.66%, 27.56%, and 361.30 mg/100g, respectively.

The TA, TS, MA, DA, SS, and Vc of 243 cultivars were plotted individually in a histogram of 10 bins. As shown in Figure 7, the bins in the histograms of TA, TS, MA, DA, SS, and Vc that had the highest numbers of cultivars ranged from 0.586% to 0.724%, 20.85% to 22.93%, 10.356% to 12.959%, 8.164% to 10.205%, 24.8% to 27.4%, and 302.934 to 370.445mg/100g, respectively.

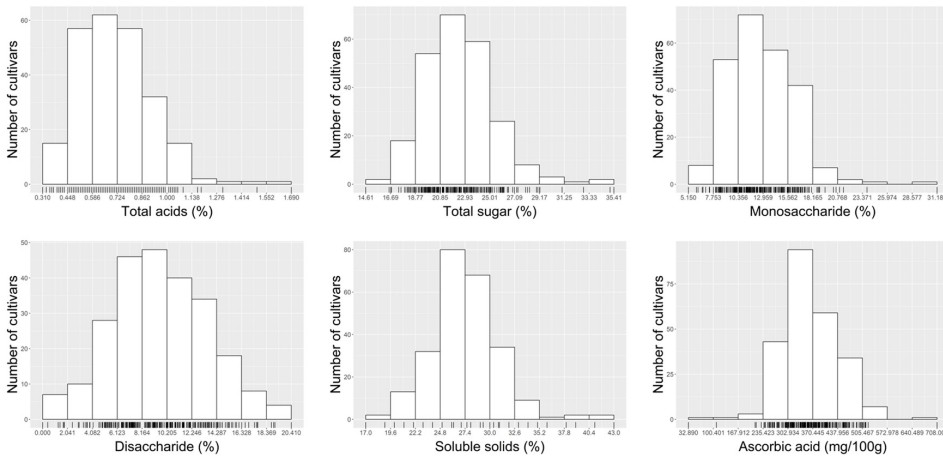

**Figure 7.** Histogram distribution of fruit chemical traits of 243 Chinese jujube cultivars.

*3.5. Correlation Analysis between Genome Sizes and Fruit Traits*

In order to study the relationships between genome size and fruit traits, 10 fruits traits, including four morphological and six chemical traits of these 243 Chinese jujube cultivars were analyzed with Pearson product–moment correlation. The results showed that a positive correlation relationship ($p < 0.05$) exists between genome size and any one of fruit size, fruit vertical dimension, fruit horizontal diameter, and fruit weight (Figure 8), with the Pearson correlation coefficients being 0.25, 0.15, 0.21, and 0.25, respectively. It is obvious that the correlation relationships between genome size and these morphological traits were generally more significant than those between genome size and morphological traits.

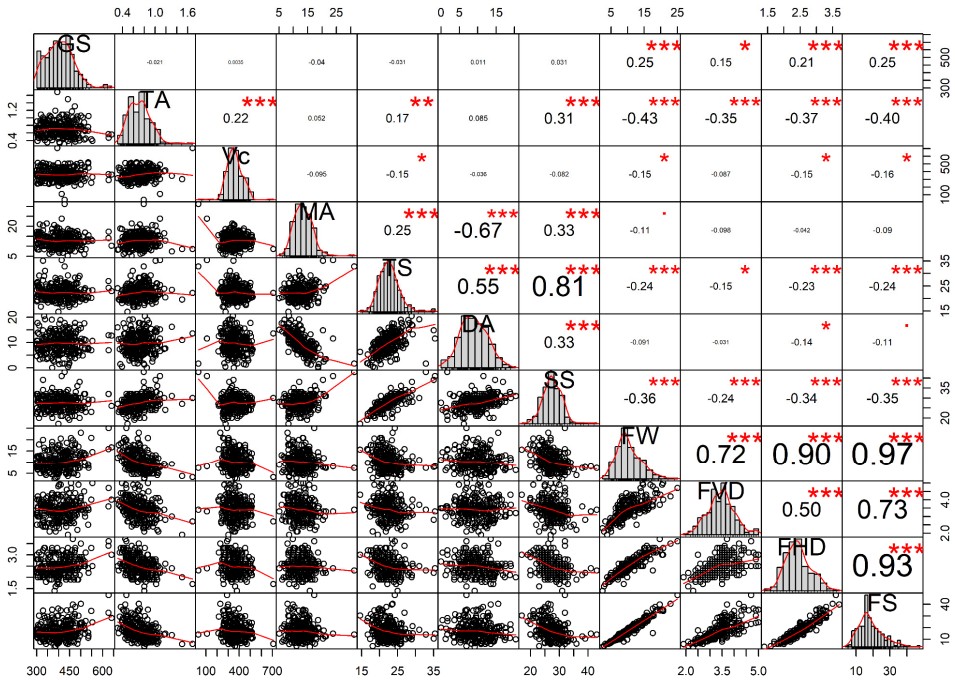

**Figure 8.** Correlation analyses between genome size and multiple fruit traits. GS: genome size; TA: total acids; Vc: ascorbic acids; MA: monosaccharides; TS: total sugars; DA: disaccharides; SS: soluble solids; FW: fruit weight; FVD: fruit vertical diameter; FHD: fruit horizontal diameters; FS: fruit sizes; The *x* and *y* axes for each plot is dependent on where the plot is. For example, the *y* axes for all plots in the first column are all GS, and the *x* axes for all plots in the bottom row are all FS. Therefore, the bottom plot in the first column has a *y*-axis of GS and an *x*-axis of FS. Asterisks *, **, and *** represent the statistical significance differences of *p*-values < 0.05, 0.01, and 0.005 (Student's *t*-test), respectively.

*3.6. Genome Size Variation in Sour Jujube*

Sour jujube exhibited a relatively small variation in genome size compared to Chinese jujube. The estimated sour jujube genome sizes in different genotypes ranged from 346.93 Mb to 489.44 Mb (Table 2). The genotype with the largest genome size in sour jujube is Xingtai0608, whose genome size is 1.4-fold of the smallest one, called Xingtai0610. The average value of genome size in sour jujube is 423.55 Mb.

**Table 2.** The genome sizes of the 81 genotypes of sour jujube.

| Genotypes | Genome Size (Mb) | | Genotypes | Genome Size (Mb) | | Genotypes | Genome Size (Mb) | |
|---|---|---|---|---|---|---|---|---|
| | Mean | SD | | Mean | SD | | Mean | SD |
| Xingtai0608 | 346.93 | 9.60 | Suanzao28 | 410.09 | 4.27 | ZiA | 434.95 | 21.93 |
| Suanzao10 | 360.30 | 3.99 | Xingtai0604 | 412.21 | 1.12 | Xingtai0614 | 436.24 | 4.22 |
| Suanzao37 | 360.78 | 8.90 | Xingtai0613 | 412.34 | 2.90 | ZiU | 439.00 | 4.08 |
| Xingtai0601 | 374.83 | 4.05 | Suanzao49 | 416.26 | 6.53 | Zi16 | 439.11 | 7.47 |
| Zi2 | 380.06 | 4.98 | Zi3 | 416.38 | 7.87 | Beikedi2 | 439.36 | 5.26 |
| Suanzao36 | 380.06 | 5.42 | Xingtai0634 | 419.24 | 3.28 | Xingtai10 | 439.84 | 17.91 |
| Xingtai0605 | 380.56 | 0.12 | ZiT | 419.58 | 5.44 | Xingtai0629 | 439.88 | 7.46 |
| Suanzao17 | 383.26 | 11.49 | Xingtai6 | 420.33 | 2.02 | Zi25 | 441.62 | 5.23 |
| Xingtai0619 | 383.70 | 1.34 | Zi4 | 421.39 | 4.69 | Taigudasuanzao | 441.63 | 8.73 |
| ZiY | 383.97 | 9.23 | Xianxiansuanzao | 422.16 | 6.41 | Suanzao40 | 443.34 | 1.76 |
| Xingtai0602 | 384.15 | 5.64 | ZiZ | 422.34 | 12.72 | Beikedi1 | 447.27 | 4.57 |
| Xingtai0618 | 385.63 | 4.80 | Zi23 | 423.55 | 5.87 | GaoVcsuanzao | 451.89 | 4.97 |
| Suanzao21 | 389.51 | 4.09 | ZiX | 424.14 | 16.42 | Suanzao11 | 456.24 | 7.06 |
| Beikedi9 | 394.09 | 3.73 | Suanzao32 | 424.18 | 3.94 | Suanzao3 | 457.33 | 1.59 |
| Zi28 | 395.12 | 1.41 | Suanzao4 | 424.96 | 7.30 | Xingtai17 | 458.94 | 3.86 |
| Suanzao27 | 398.72 | 3.52 | Suanzao45 | 426.95 | 0.80 | Suanzao6 | 458.98 | 0.39 |
| Suanzao41 | 398.91 | 2.63 | Suanzao30 | 427.61 | 1.71 | Suanzao1 | 459.11 | 3.32 |
| Jiaochengtiansuanzao | 398.93 | 3.67 | Suanzao44 | 428.25 | 9.48 | Suanzaowang | 459.47 | 8.66 |
| Xingtai0603 | 398.95 | 2.65 | Suanzao9 | 428.43 | 2.52 | Beiqi6 | 459.67 | 6.00 |
| ZiO | 400.97 | 3.77 | Wumingsuanzao | 429.79 | 0.68 | Xingtai27 | 465.21 | 6.91 |
| Zi24 | 403.92 | 9.60 | Suanzao12 | 430.30 | 1.83 | Suanzao20 | 466.38 | 6.59 |
| Zi15 | 405.53 | 9.89 | ZiV | 430.66 | 2.54 | Suanzao24 | 474.38 | 3.47 |
| Xingtai0648 | 406.17 | 3.94 | Suanzao8 | 431.31 | 2.82 | Xingtai0609 | 479.91 | 7.97 |
| Suanzao23 | 406.63 | 8.65 | Suanzao34 | 431.79 | 2.55 | Xingtai16 | 482.34 | 4.22 |
| Beikedi3 | 406.69 | 3.94 | Suanzao14 | 431.87 | 2.78 | Suanzao25 | 483.04 | 7.81 |
| ZiH | 407.28 | 5.05 | Suanzao47 | 432.51 | 1.86 | Suanzao43 | 486.04 | 4.89 |
| Suanzao31 | 408.87 | 3.12 | Suanzao48 | 433.84 | 5.36 | Xingtai0610 | 489.44 | 1.29 |

A distribution histogram was made based on the genome sizes of 81 cultivars in sour jujube (Figure 9). The numbers of genotypes that fell into the 10 bins were 3, 1, 9, 8, 11, 20, 12, 9, 3, and 5. The results revealed that the bin with more sour jujube genotypes included is the one that ranges from 418.185 to 446.687 Mb.

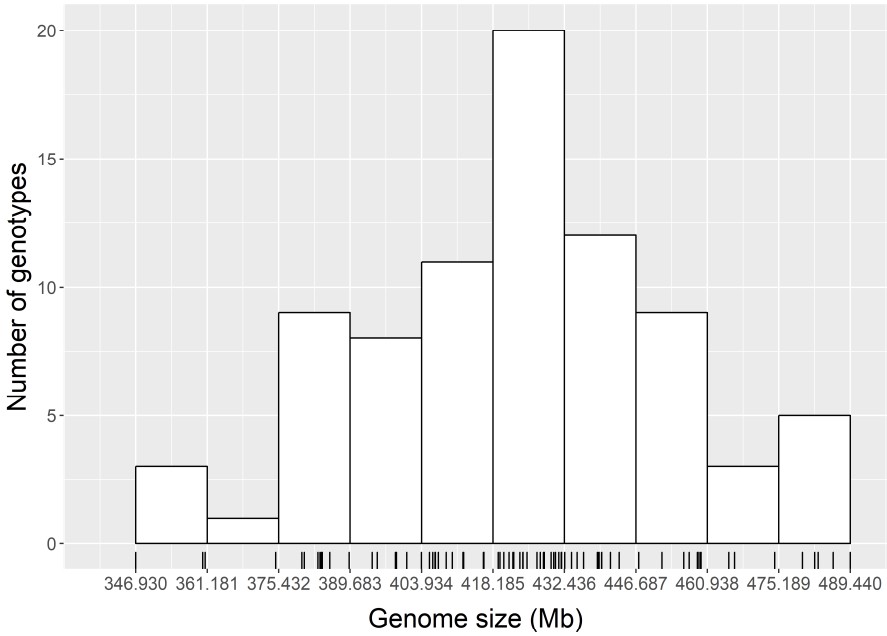

**Figure 9.** Histogram distribution of genome sizes of 81 genotypes of sour jujube.

*3.7. Comparison of Genome Sizes between Chinese Jujube and Sour Jujube*

The median and average values of genome sizes were 404.68 Mb and 408.54 Mb for Chinese jujube and 424.18 Mb and 423.55 Mb for sour jujube, respectively (Figure 10). The overall genome size of Chinese jujube genome is smaller than that of sour jujube. For Chinese jujube, there were three extreme outliers (triploids), "Zanxindazao", "Jinzandazao", and "Zanhuangdazao", whose genome sizes were 626.21 Mb, 627.13 Mb, and 640.94 Mb [19], respectively. After removing the three triploids, the average genome size of the Chinese jujube is 406.29 Mb.

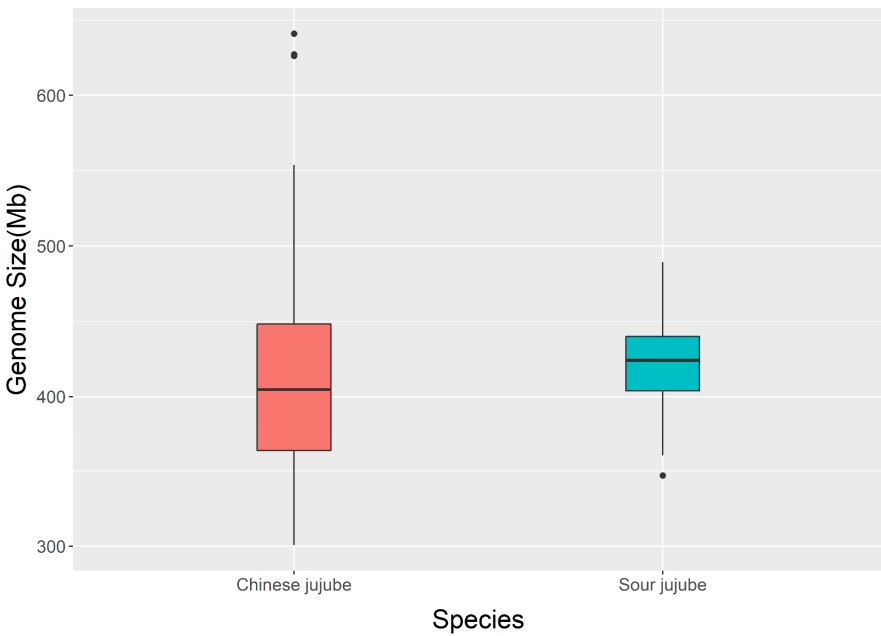

**Figure 10.** Boxplot of genome sizes in Chinese jujube and sour jujube. Each bar represents the median value of the genome size of either Chinese jujube or sour jujube.

Though the median and average genome sizes of sour jujube were larger than those of Chinese jujube, we noticed that the sour jujube genotypes we used were mostly from Hebei and Shaanxi

provinces. To increase the comparability, we also compared the genome sizes of Chinese jujube and sour jujube using the cultivars or genotypes from these two provinces. The results showed that the average genome size of sour jujube was still slightly larger than that of Chinese jujube (Figure 11).

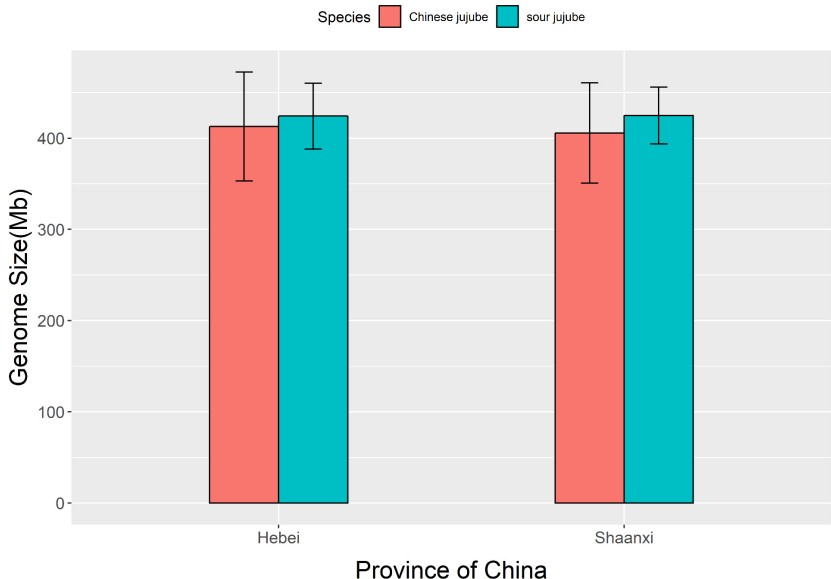

**Figure 11.** Comparison of average genome sizes of Chinese jujube and sour jujube from Hebei and Shaanxi provinces.

## 4. Discussion

### 4.1. The Genome Size of Chinese Jujube and Sour Jujube

In this study, the genome sizes of 301 Chinese jujube cultivars and 81 sour jujube genotypes were measured using flow cytometry, and considerable variations were found in the cultivars/genotypes of both species. The estimated genome sizes of Chinese jujube and sour jujube ranged from 300.77 Mb to 640.94 Mb (408.54 Mb on average) and 346.93 Mb to 489.44 Mb (423.55 Mb on average), respectively. When the three triploid cultivars of Chinese jujube were excluded, the genome sizes of Chinese jujube ranged from 300.77 MB to 553.79 Mb (406.29 Mb on average). These new data on the genome sizes of both Chinese jujube and sour jujube are larger than the ranges of Chinese jujube genome size in previous reports (337.43 Mb to 379.49 Mb), after investigating nine cultivars of Chinese jujube [23]. Given the large variation in genome sizes of both species, it is essential to investigate a large number cultivars/genotypes before a more accurate estimation of their average genome sizes can be achieved.

Prior to this study, flow cytometry has been used to determine the genome sizes of several Chinese jujube cultivars (nine) and sour jujube genotypes (four) [23,24]. Compared with these previous studies, we used two sequenced plants, *Populus trichocarpa* and *Ziziphus jujuba* "Dongzao", as references. *P. trichocarpa* was used as baseline, while "Dongzao" was used as a test cultivar to reduce systematic error. By comparing the genome size of the same cultivars/genotypes with the previous results [23,24], it was found that there was only a 3.09% difference in "Xianxiansuanzao" (sour jujube) and a 1.60% difference in "Jinzao" (Chinese jujube). This indicates that the test results are close when using different reference plants and different methods for lysates, suggesting the stability of FCM technology. However, the genome size of "Dongzao" is 444 Mb in this experiment, but previous studies have shown the genome sizes of "Dongzao" to be 352.41 Mb and 393.60 Mb [23,24]. The possible reasons for this difference might be the different origins of plant materials used (in vitro, in vivo, or from different regions). In this study, the genome sizes of "Dongzao" from Gansu, Heibei, Hebei, and Shandong are 306.97 Mb, 420.89 Mb, 384.66 Mb, and 474.57 Mb, respectively. This clearly indicates that the "Dongzao" of different origins have different sizes of genomes. A previous study using an RAPD

(random amplified polymorphic DNA) marker also showed that "Dongzao" from different habitats had great genetic diversity [25]. Therefore, caution should be taken when cultivars/genotypes are selected for genome sequencing and other genome-based studies.

Upon the completion of the genome sequencing of Chinese jujube "Dongzao" in 2014 [21] and "Junzao" in 2016 [26], it was found that the genomes of these two cultivars differed in size by approximately 86.5 Mb. This fact indicates that there is great variation in the genome sizes within the cultivars of Chinese jujube. Huang et al [26] speculated that the 86.5 Mb difference is partially due to the recent insertions of transposable elements (TEs) in the "Dongzao" genome, because non-genic DNA in the plant genome mainly consists of transposable elements whose proliferation can drive genome expansion. The comparative genome analysis of "Dongzao" and "Junzao" suggests that the larger genome of "Junzao" is partly due to the insertions of duplicate TEs [26]. In addition, all angiosperms are paleopolyploids, because their genomes underwent whole genome duplication (WGD) at least one time in their evolutionary history [27]. The legacy of past polyploidization was masked in many species by the "diploidization" process, which includes chromosomal re-arrangement, redundant gene loss, and neofunctionalization [28–31]. Presumably, the evolution of "Dongzao" and "Junzao" have been subject to different diploidization processes that have resulted in the difference in their genome sizes.

### 4.2. Association of Genome Size with Fruit Phenotype in Chinese Jujube

The phenotypes and ecological characteristics of various plant species are frequently associated with the variation in genome sizes [32]. Therefore, genome sizes become an indispensable attribute used to assist in the analysis of plant evolution and biodiversity [33]. A previous report found that genome size is a significant trait that can affect biomass growth under different nutrient regimes, influencing plant community composition and ecosystem dynamics [8]. It was found that there are some correlative relationships between plant genome sizes and many traits, possible because both have been shaped by climate changes and environmental conditions. As reported, seed mass and specific leaf area minimum generation time are positively correlated to genome sizes, while growth rate, seed number, and water and nutrient use efficiency are negatively correlated to genome sizes [3]. It is also reported that plant genome sizes have some impact on plant invasiveness [34]. However, contradicting evidence has been manifested for some species like *Phalaris arundinacea*, where genome sizes bear little correlation with either growth rates or invasive and native range accessions [35]. These results suggest that the relationships between genome sizes and phenotypic traits vary with species.

In this study, we found that there was a significant, weakly positive correlation between genome size and fruit size, fruit vertical diameter, fruit horizontal diameter, and fruit weight in Chinese jujube. It was found that there is a typical correlation between genome size and cell volume [32,36]. Owing to its effects on cell size-related parameters and cell division rates, genome size evidently affects both size- and rate-dependent traits, which can cause changes in the sizes of fruit. This lays some basis for the observed fruit size and related changes in this study, to a certain extent. However, the observed correlations in statistics may or may not indicate that there is a causal relationship between them.

### 4.3. Genome Downsizing during Domestication of Chinese Jujube

In the process of domestication, plant genomes are usually subjected to three types of changes, i.e., contraction, expansion, and steady state. The steady state in the genome is incurred primarily in ferns compared to angiosperms [37]. Genome expansion has been reported in many plant species [38], such as *Arabidopsis* [39], and grape [40]. Genome downsizing has been also found in many plant species. Whole-genome sequencing of wild species and cultivars of pepper have revealed that the genome size was reduced 131 Mb during domestication [41]. Similarly, the genome sizes of several cultivars of olive have contracted nearly 10% [42]. Our study uncovered that the evolution of Chinese jujube (average genome size of 406.29 Mb without triploids) from sour jujube (average genome size of 423.55 Mb) appears to have followed the trajectory of genome contraction.

In addition, there were more nucleotides that were inevitably subject to more mutations per genome when the plants suffered from greater external stress or environmental changes [43], which may indicate that sour jujube (wild species) has experienced stronger environmental harshness than Chinese jujube. Simonin and Roddy [44] reveal that the contraction of angiosperm genomes was incurred in response to harsh environmental conditions. The dramatic contraction of the angiosperm genomes has led to the packing of more veins and stomata into their leaves, effectively bringing actual primary productivity closer to its maximum potential, thereby improving their environmental adaptability. In this study, the average genome size of Chinese jujube from Liaoning Province, the northernmost province for Chinese jujube, is smaller, which is evidently an example that suggests that the cultivar in Liaoning has been experiencing an adaptation to the cold environment.

### 4.4. The Possible Mechanism of Large Genome Size Formation in Chinese Jujube

Polyploidization is regarded as one important driving force for plant evolution [45], as it can provide a large number of raw materials for biodiversification, speciation, and new genetic combinations that favor evolution [46]. The main path of polyploidization is whole-genome duplication (WGD). With the completion of a large number of plant genome sequencing projects, it has been found that WGD events have occurred in most plant evolutionary processes [21,47–49]. The evolutionary process of polyploidization can be roughly divided into three stages [50]. First, the plant chromosomes duplicate under an environmental stressor. Secondly, plants with a duplicated genome adapt to the stress caused by the larger duplicated genome, in which both genetic and epigenetic changes occur to drive structural and functional reorganization. During this period, gene losses and neofunctionalization also occur. Thirdly, the plants eventually undergo diploidization to produce a new diploid with a relatively stable phenotype.

In this study, the genome sizes of the three known triploid Chinese jujube cultivars were all over 600 Mb (626.21 to 640.94 Mb, approximately 1.5 times the average size of diploids), but some other cultivars also showed quite a large genome size (more than 500 Mb, but less than 600 Mb). These cultivars include "Jishanchangzao", "Shandonglizao", "Yanchuandieyazao", "Xinledazao", "Chaoyangjianjianzao", "Yucijiuyueqing", "Weihaijinsi2", "Hubeiyuanzao", "Guoxingpingguozhuang", "Puyangtangzao", "Xiaxianyuancuizao", "Xiangfenyuanzao", "Shenxianchuanganzao", "Ningxiada hongzao", "Taiguhuluzao", "Shaoguanbaizao", "Lichengdamazao", "Shaizao", "Linyilizao", and "Puyangsanbianchou" have been reported to be diploid in previous reports [19]. This discovery implicates that different diploid cultivars could have different genome sizes. As mentioned previously, the "Juncao" genome expanded in part owing to duplicated TEs [26]; this can be one cause for the observed differences in different Chinese jujube cultivars. In addition, all angiosperms, including all the Chinese jujube cultivars, underwent at least one whole genome duplication, which was followed by different post-polyploidization diploidization processes, presumably as an adaptation to various environmental conditions or climate types, leading to different genome sizes. These two suppositions can be studied further once the genome sequences of more Chinese jujube cultivars become available in the future.

## 5. Conclusions

This is the first large-scale extensive survey of genome size variation in both Chinese jujube and sour jujube. The results manifested a high variation in genome sizes within both species. The average genome size of Chinese jujube is slightly smaller than that of sour jujube, supporting that genome contraction may have been initiated during the domestication of Chinese jujube. The cultivars from the periphery of the traditional distribution area appear to contract more, supporting that environmental conditions were probably the major driving force. Our results suggest that the species with a large distribution area spanning multiple climate types usually have greater variation in genome sizes and genetic variation in various traits, which should be ascertained and used in genetic conservation, as well as in conventional and molecular breeding.

**Supplementary Materials:** The following are available online at http://www.mdpi.com/1999-4907/10/5/460/s1, Table S1: Basic information for 301 Cultivars of Chinese jujube and 81 genotypes of sour jujube, Table S2: Major climate conditions and climate types in various provinces of China, Table S3: Genome size of 301 cultivars of Chinese jujube, Table S4: Fruit traits and quality of the 243 cultivars of Chinese jujube.

**Author Contributions:** L.W., M.L., P.L. and J.Z. conceived and designed the experiments; L.W., Z.L. (Zhi Luo) and Z.L. (Zhiguo Liu) performed the experiments; L.W., W.D., H.W. and M.L. analyzed the data and wrote the paper.

**Funding:** This work was supported by the National Natural Science Foundation of China (31372029), the Natural Science Foundation of Hebei, China (C2017204114), the Hebei Province Talent Training Project (A201501026), the Science and Technology Plan Project of Hebei Province (16226313D-1), 2017 Provincial Graduate Student Innovation Project of Hebei (CXZZBS2017072).

**Acknowledgments:** We thank Dengke Li and Xiaofang Xue for their help with the jujube sample collection, as well as Xingjuan Zheng and Lihui Zuo for assistance in flow cytometry analysis.

**Conflicts of Interest:** The authors declare no conflict of interest.

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
