# Peer review of "Genome Size Variation within Species of Chinese Jujube (Ziziphus jujuba Mill.) and Its Wild Ancestor Sour Jujube (Z. acidojujuba Cheng et Liu)"

_forests, doi:10.3390/f10050460_

Round 1

Reviewer 1 Report

The paper describes the genome size variation within species of Chinese jujube (Ziziphus jujuba Mill.) and its wild ancestor sour jujube (Z. acidojujuba Cheng et Liu). The findings of this paper could rich the knowledge’s in genome size variation, breeding programs and cultivation of these species.

Overall, the manuscript is basically written clearly. The results are clearly presented and the discussion is basically reasonable and sound. However, in my opinion the paper could be improved.

Subtitle 2.3. Chinese jujube fruits trait measurement can be divided into two separate chapters: Morphometric analysis and Chemical analysis. In addition, chemical analysis should be better described. Furthermore, morphological and chemical diversity should be analysed and described as well. The only analysis with fruit data was correlation analysis between genome sizes and fruit traits. I think that authors could perform other additional analysis with fruit traits, i.e. morphometric and chemical analysis (see Food Technol. Biotechnol. 54 (2) 189 – 199, 2016; and Sumarski List 142 (1-2) 19-32, 2018). One additional subtitle describing fruit traits analysis in the sections Results and Discussion should be included.

Author Response

The paper describes the genome size variation within species of Chinese jujube (Ziziphus jujuba Mill.) and its wild ancestor sour jujube (Z. acidojujuba Cheng et Liu). The findings of this paper could rich the knowledge’s in genome size variation, breeding programs and cultivation of these species. Overall, the manuscript is basically written clearly. The results are clearly presented and the discussion is basically reasonable and sound. However, in my opinion the paper could be improved.

< Response> Thank you very much for your favorable comment.

Subtitle2.3. Chinese jujube fruits trait measurement can be divided into two separate chapters: Morphometric analysis and Chemical analysis. In addition, chemical analysis should be better described. Furthermore, morphological and chemical diversity should be analysed and described as well. The only analysis with fruit data was correlation analysis between genome sizes and fruit traits. I think that authors could perform other additional analysis with fruit traits, i.e. morphometric and chemical analysis (see Food Technol. Biotechnol. 54 (2) 189 – 199, 2016; and Sumarski List 142 (1-2) 19-32, 2018). One additional subtitle describing fruit traits analysis in the sections Results and Discussion should be included.

< Response > Thanks very much for your suggestions. In the revised manuscript, subtitle 2.3, Chinese jujube fruits trait measurement, has been split into two separate titles named “Fruit morphological analysis” and “Fruit chemical analysis”. In addition, the morphological and chemical variations in 243 cultivars of Chinese jujube were analyzed, and one subtitle namely “3.4. Morphological and chemical variation in fruit among cultivars of Chinese jujube” has been added in the Results to describe the results of fruit trait related analyses.

We read articles “Morphological Characterization and Chemical Composition of Fruits of the Traditional Croatian Chestnut Variety Lovran Marron” and “Diversity and structure of croatian continental and alpine-dinaric populations of grey alder (Alnus incana /L./ Moench subsp. incana); Isolation by distance and environment explains phenotypic divergence” carefully. We will conduct a more detailed analysis in fruit morphological and chemical traits in the future, and the information you provided is of great helpful to us. In this study, the subject of research is primarily focuses on the variation in the genome size of Chinese jujube and sour jujube. We mainly want to know whether there is a certain association between the genome size and the fruit traits, which is shown in discussion section under the Subtitle 4.2.

In addition, we have re-check the full manuscript and improved it.

Reviewer 2 Report

The authors examined genome size of 301 cultivars of Chinese jujube and 81 genotypes of sour jujube by flow cytometry. They conducted correlation analyses of genome size with phenotypes and geographical source of the cultivars. While providing genome size information of the cultivars and correlation of genome size with phenotypes bear scientific merits, it is difficult to justify the significance of studying correlation of genome size with geographical source of the cultivars, because the authors did not mention how these cultivars were generated in the first place (were they created locally with local wild species or were they introduced from other places?). Without such information, it is difficult to interpret the results and discuss domestication-related topics. Another main concern is the English writing in the manuscript. Besides grammatical errors (a few examples are listed below), the statistical analysis method lacks details (what model was used? What kind of test were used for significance analysis? Etc.), and the discussion is weak. • 20.93% and 24.69% are not majority. • For references Z. jujuba 'Dongzao and Populus trichocarpa, how many repeats were included? • No need to spell out genus name all the time. • Lines 76 and 77: double check “ grown under in the XXX” • Lines 86-88: double check the sentence. • Line 98: correct “Each sample should collected” • Line 109: “ascorbic acid (Vc),those”: missing space • Line 125: format error in Figure 1 legend. • Line 154: Figure 3 legend “boxplot”: capitalize “B” • Figure 6: fruit traits are not indicted in the table. • Line 210 and 211: correct the sentence • Lines 225~227: I think this sentence does not convey what you wanted to mean. • Line 232-233: correct the sentence • Line 250: The authors said “effect of genome size on fruit phenotype”. I think “association” of genome size and fruit phenotype is more accurate. • The authors suggested that “the selection of different provenances may lead to different conclusions” for the reference Dongzao. In this case, the reference is not a suitable reference. Plus, the source of the reference was not provided in the manuscript. • “Huang et al [26] think that those 83Mb difference partially due to the recent insertions of transposable elements in the ‘Dongzao’.”: this sentence is grammatically incorrect. And what is the evidence that led to the suggestion of the role of TE in the 83Mb difference? • Line 256: “passible” is not the correct word.

Author Response

The authors examined genome size of 301 cultivars of Chinese jujube and 81 genotypes of sour jujube by flow cytometry. They conducted correlation analyses of genome size with phenotypes and geographical source of the cultivars. While providing genome size information of the cultivars and correlation of genome size with phenotypes bear scientific merits, it is difficult to justify the significance of studying correlation of genome size with geographical source of the cultivars, because the authors did not mention how these cultivars were generated in the first place (were they created locally with local wild species or were they introduced from other places?). Without such information, it is difficult to interpret the results and discuss domestication-related topics.

< Response > Thank you very much for raising such a critical problem. I am very sorry for not expressing clearly in the manuscript. The whole paragraph has been rewritten in the revised manuscript, and we listed the original sources of all samples in Supplementary Table 1. In fact, the Chinese jujube cultivars (local cultivars) conserved at National Jujube Germplasm Repository were originally collected from different provinces of China where they have been cultivated for hundreds or thousands of years, and all sour jujube genotypes conserved at the Agricultural Experimental Station, Hebei Agricultural University were collected from their wild habitats in various provinces of China.

Another main concern is the English writing in the manuscript. Besides grammatical errors (a few examples are listed below), the statistical analysis method lacks details (what model was used? What kind of test were used for significance analysis? Etc.), and the discussion is weak.

< Response > We have looked through the whole manuscript and revised it thoroughly. The statistical analysis method has been supplemented in the revised manuscript. In addition, We made some changes to the discussion section in the revised manuscript.

•20.93% and 24.69% are not majority.

< Response > “majority” has been changed into highest number of cultivar/genotype” or something that is similar.

•For references Z. jujuba 'Dongzao and Populus trichocarpa, how many repeats were included?

< Response > The repeat number (three) has been added in the revised manuscript.

•No need to spell out genus name all the time.

< Response > It has been modified in the revised manuscript.

•Lines 76 and 77: double check “grown under in the XXX”

< Response > Thank you for your reminder! The whole sentence has been replaced with “301 cultivars of Chinese jujube and 81 genotypes of sour jujube were sampled at National Jujube Germplasm Repository (NJGR), Taigu county, Shanxi Province, China and the Agricultural Experiment Station, Hebei Agricultural University (HAU), Baoding, Hebei Province, China, respectively.”

•Lines 86-88: double check the sentence.

< Response > We apologize for this error. It has been changed into “Populus trichocarpa was used as the baseline reference for measuring genome size of Chinese jujube and sour jujube in order to ensure the reliability of the measured results.”

•Line 98: correct “Each sample should collected”

< Response > “Each sample should collected” has been changed into Each sample should be collected”.

•Line 109: “ascorbic acid (Vc), those”: missing space

< Response > It has been modified in the revised manuscript.

•Line 125: format error in Figure 1 legend.

< Response > It has been modified in the revised manuscript.

•Line 154: Figure 3 legend “boxplot”: capitalize “B”

< Response > It has been modified in the revised manuscript. We also added more explanation to the legend.

• Figure 6: fruit traits are not indicted in the table.

< Response > We apologize for this error. It has been modified in the revised manuscript.

•Line 210 and 211: correct the sentence

< Response > “Though the median and average genome sizes are representative for comparing the genome sizes the Chinese jujube and sour jujube” has been changed into “Though the median and average genome sizes of sour jujube were larger than those of Chinese jujube”.

•Lines 225~227: I think this sentence does not convey what you wanted to mean.

< Response > We have changed the sentence (it is found that there is only a small difference between Xianxiansuanzao (sour jujube) and Jinzao (Chinese jujube), 3.09% and 1.60%, respectively) into “it was found that there is only 3.09% difference in ‘Xianxiansuanzao’ (sour jujube) and 1.60% difference in ‘Jinzao’ (Chinese jujube)”.

•Line 232-233: correct the sentence

< Response > We have changed the sentence (The reason for this distinct difference is that the plant materials subjected to the tests from different provenances (in vitro and in vivo, or different regions) into “The possible reasons for this difference might be the different origins of plant materials used (in vitro, in vivo or different regions).”

•Line 250: The authors said “effect of genome size on fruit phenotype”. I think “association” of genome size and fruit phenotype is more accurate.

< Response > It has been modified as your suggestion in revised manuscript.

•The authors suggested that “the selection of different provenances may lead to different conclusions” for the reference Dongzao. In this case, the reference is not a suitable reference. Plus, the source of the reference was not provided in the manuscript.

< Response > We have modified the sentence “suggest that the selection of different provenances may lead to different conclusions” into “This clearly indicates that the ‘Dongzao’ of different origins have different sizes of genomes”.

• “Huang et al [26] think that those 83Mb difference partially due to the recent insertions of transposable elements in the ‘Dongzao’.”: this sentence is grammatically incorrect. And what is the evidence that led to the suggestion of the role of TE in the 83Mb difference?

< Response > Thank you for your reminder. We read the article of Huang again and corrected the sentence in the revised manuscript.

•Line 256: “passible” is not the correct word.

< Response > “passible” has been changed into “possible”.

In addition, we have re-check the full manuscript and improved it.

We hope our revision could meet your request.

Thank you again!

Reviewer 3 Report

The research expands the knowledge on jujube genome. Although the research is of some interest it needs some improvement, namely on the results and their interpretation of the geographical relations, chemical traits and morphological raits with genome size. Please find some detailed suggestions in the edited MS file.

Author Response

May, 10. 2019

Dear reviewer,

Thanks very much for your kindly and constructive comments. We have tried to improve the manuscript according to your valuable comments and queries, we used the "track changes" in the modified manuscript.

The research expands the knowledge on jujube genome. Although the research is of some interest it needs some improvement, namely on the results and their interpretation of the geographical relations, chemical traits and morphological traits with genome size. Please find some detailed suggestions in the edited MS file.

< Response > Thanks very much for your positive affirmation and valuable suggestions. In the revised manuscript, we made the appropriate changes based on your comments.

Line 377-379 Please clarify and cite reference literature

< Response > Sorry for the unclearness. In the revised manuscript, we have explained the section in detail and cite more literature in the revised manuscript, hoping that our changes will help you understand what we meant better.

Line 417, 419 redundant

< Response > These sentences have been deleted in the revised manuscript.

Line 458-459 Are there documented experiences of that event in other species?

< Response > We have changed the text to the following. As afremensioned, the comparative study indicated the ‘Juncao’ genome expanded in part owing to the duplicated TEs [26], this can be one cause under the observed differences in different Chinese jujube cultivars. In addition, all angiosperms including all the Chinese jujube cultivars underwent at least one whole genome duplication that was followed by different post-polyploidization diploidization processes presumably in an adaptation to various environmental conditions or climate types, leading to different genome sizes. These two suppositions can be studied once the genome sequences of more Chinese jujube cultivars become available in the future.

Line 461 what will be the next steps? Can you say quickly?

< Response > The original sentence of “Of course, this surmise needs to be confirmed in the future research” has been change into “These two suppositions can be studied once the genome sequences of more Chinese jujube cultivars become available in the future.”. We think the next step is to get the genome sequences of a few more Chinese jujube cultivars and then conduct comparative studies on genome contraction/expansion, and/or compare the genome sequences from multiple climate types to verify the above two suppositions.

Line 470 redundant

< Response > These sentences have been deleted in the revised manuscript.

In addition, we have re-checked the full manuscript and improved it.

We hope our revision could meet your request.

Thank you very much again!

Mengjun Liu

Research Center of Chinese Jujube

Hebei Agricultural University

289, Lingyushi Street

Baoding,071001, China

Phone: +8613932262298

Email: kjliu@hebau.edu.cn

Reviewer 4 Report

The evolutive mechanism related to expansion and contraction of genomes are clearly represented by this example on jujuba. Perhaps, a table with main climatic or other driving factors per province or region could be included, in order the reader can understand the climatic variation in China.

Please find some minor suggestion in the text.

Author Response

May 10. 2019

Dear reviewer,

Thanks very much for your kindly and constructive comments. We have tried to improve the manuscript according to your valuable comments and queries, we used the "track changes" in the modified manuscript.

The evolutive mechanism related to expansion and contraction of genomes are clearly represented by this example on jujuba. Perhaps, a table with main climatic or other driving factors per province or region could be included, in order the reader can understand the climatic variation in China.

Please find some minor suggestion in the text.

< Response > Thanks very much for your suggestions. In the revised manuscript, we made the appropriate changes based on your suggestions. We have added Table S2 that contains the climatic conditions and climate types of major Chinese provinces in the revised manuscript, hoping that this material will help interested readers to delve into it for more valuable information.

Line 92,95,149, “in vitro culture tissues”, Explain this. Did you use samples from tissue culture lines?

< Response > The tissue culture lines include only Dongzao and Populus trichocarpa whose genomes have been sequenced. The samples of 301 Chinese jujube cultivars and 91 sour jujube genotype which were used in this study are not included in tissue culture lines.

Line 158 Bibliographic reference?

< Response > Sorry for the inconvenience. We have added the references.

Line 275 The figure needs further explanation of what the axis represent. As it is it's very difficult to interpret.

< Response > The figure is made by R package called Performance Analytics. This figure is a classic picture of correlation analysis. The x and y axes are labeled by the words (GS, TA, Vc, MA, TS, …, FHD and FS) as shown in the diagonal. Take the last picture in the first column of the figure as an example, this figure represents the correlation between genome size (GS) and fruit size (FS), the horizontal axis (x) represents the genome size and the vertical axis (y) represents the fruit size. We have added an explanation to the legend of Figure 8 to help readers figure out a and y axes for each plot.

Line 377 “non-genetic DNA” not clear?

< Response > “non-genetic DNA “has been changed to “non-genic DNA”. It refers to the regions that do not contain genes.

Line 378 “some scholars believe”, Please explain this further and provide more references.

< Response > We have removed the phrase of “some scholars believe”, and then added more references to explain our early statements in more detail in the revised manuscript, hoping that our explanation has elucidated it more clear. Briefly, based on the earlier study by Huang et al, the larger genome of ‘Junzao’ is partly caused by the inserions of the duplication of TEs. In addition, all angiosperms underwent at least one whole genome duplication (WGD) followed by different post-ployploidrization diploidization processes, which led to the differences in the genome sizes of different diploid cultivars. We have added a paragraph to explain this scenario.

Line 383 Discussion of geographic variation?

< Response > In this section, we only discuss the relationship between the genome and the fruit phenotype, and do not involve geographical variation. However, your suggestion may provide new ideas for our next research, and we will conduct special research on this in future research. Thank you again.

In addition, we have re-checked the full manuscript and improved it.

We hope our revision could meet your request.

Thank you again!

Mengjun Liu

Research Center of Chinese Jujube

Hebei Agricultural University

289, Lingyushi Street

Baoding,071001, China

Phone: +8613932262298

Email: kjliu@hebau.edu.cn